# Algorithmics, Possibilities and Limits of Ordinal Pattern Based Entropies

**DOI:** 10.3390/e21060547

**Published:** 2019-05-29

**Authors:** Albert B. Piek, Inga Stolz, Karsten Keller

**Affiliations:** 1Institute of Mathematics, University of Lübeck, D-23562 Lübeck, Germany; 2Graduate School for Computing in Medicine and Life Sciences, University of Lübeck, D-23562 Lübeck, Germany; 3Department of Mathematics, The University of Flensburg, D-24943 Flensburg, Germany

**Keywords:** symbolic analysis, ordinal patterns, Permutation entropy, conditional entropy of ordinal patterns, Kolmogorov-Sinai entropy, algorithmic complexity

## Abstract

The study of nonlinear and possibly chaotic time-dependent systems involves long-term data acquisition or high sample rates. The resulting big data is valuable in order to provide useful insights into long-term dynamics. However, efficient and robust algorithms are required that can analyze long time series without decomposing the data into smaller series. Here symbolic-based analysis techniques that regard the dependence of data points are of some special interest. Such techniques are often prone to capacity or, on the contrary, to undersampling problems if the chosen parameters are too large. In this paper we present and apply algorithms of the relatively new ordinal symbolic approach. These algorithms use overlapping information and binary number representation, whilst being fast in the sense of algorithmic complexity, and allow, to the best of our knowledge, larger parameters than comparable methods currently used. We exploit the achieved large parameter range to investigate the limits of entropy measures based on ordinal symbolics. Moreover, we discuss data simulations from this viewpoint.

## 1. Introduction

Symbolic-based analysis techniques are efficient and robust research tools in the study of non-linear and possibly chaotic time-dependent systems. Initially, an experimental time series $x_{0},x_{1},\ldots ,x_{N}$ with *N* in the natural numbers N is decoded into a sequence of symbols and, if needed, successive symbols are conflated into symbol words of length m∈N. Subsequently, these symbol sequences or symbol word sequences are analyzed mainly by considering symbol (word) distributions or quantifiers based on the distributions, such as entropies. Anomalies in the symbol data or changes in the entropy can be used to detect temporal characteristics in the data. As a result of long-term data acquisition and high sample rates, the study of long-time series is becoming increasingly important in order to provide useful insights into long-term dynamics [1]. Efficient and robust algorithms are required that can analyze long time-series without decomposing the data into smaller series.

In this paper, we are especially interested in the relatively new ordinal symbolic approach which goes back to the innovative works of Bandt and Pompe [2] and Bandt et al. [3]. In the symbolization process, the ordinal approach regards the dependence of d+1 equidistant data points resulting in ordinal patterns of order d∈N. The ordinal approach is applied in many research areas to identify interesting temporal patterns that are hidden in the data. Application examples, techniques and further details are listed in the review papers of Kurths et al. [4], Daw et al. [5] and Zanin et al. [6]. In addition, see the contributions to the special topic “Recent Progress in Symbolic Dynamics and Permutation Complexity. Ten Years of Permutation Entropy” of The European Physical Journal [7] and to the special issue “Symbolic Entropy Analysis and Its Applications” of Entropy [1]. The significant demand for literature on the ordinal idea is due to the fact that the method is easy to interpret and efficient to apply. However, large values of *N*, *d* and word lengths *m* lead to capacity or, on the contrary if *N* is reduced, undersampling problems. Here *m* is the number of successive ordinal patterns forming the words considered in the analysis.

This paper covers two main objectives: efficient algorithms for determining ordinal pattern (word) distributions and applicability of ordinal methods. The algorithms use overlapping information and binary representations of ordinal patterns and therefore realize an efficient determination of different entropy measures including and generalizing permutation entropies [2]. To the best of our knowledge, our algorithm not only allows larger parameters *N*, *d* and *m* than methods presently used but also compute a whole series of entropies based on ordinal words of different length and pattern order. A discussion of algorithmic complexity and runtime of the given algorithms in comparison to other ones is provided. Our algorithms are particularly useful for getting deeper insights into the general applicability of ordinal methods, which is the second objective of the paper. We discuss limits of estimating the complexity of finite time series and underlying systems on the base of ordinal patterns and words. There are different restrictions. First of all, if a system provides a large variety of ordinal patterns or ordinal pattern words with sufficiently large *d* or *m*, then a time series must be extremely long to represent the variety. This naturally bounds useful orders *d* and word lengths *m* (see e.g., [8,9] for further information). Even in the case that extremely long series and large *d* and *m* would be accessible, complexity estimation can be limited. Let us shortly explain reasons for that.

Central complexity measures for dynamical systems are related to the Kolmogorov-Sinai entropy being mathematically well-founded but being not easy from the computational and estimation viewpoint. There are different ways of approximating Kolmogorov-Sinai entropy on the base of ordinal pattern (word) distributions for sufficiently large *d* or *m* (see Bandt et al. [3] and Gutjahr and Keller [10] for ordinal pattern distributions and Keller et al. [11] for ordinal word distributions). Since there are systems with arbitrarily large, even infinite, Kolmogorov-Sinai entropy, there is no *d* or *m* which is large enough for reaching the Kolmogorov-Sinai entropies for all systems. Moreover, in case that data considered are directly an orbit of a time-discrete dynamical system (representing the system), calculation precision bounds the number of accessible ordinal patterns and words. The reason is that, independent from *d* and *m*, there are not more patterns and words than possible values of the system in the precision decided. The aspect of calculation precision is also interesting for data simulation including the problem of getting periodicities by rounding.

Many researchers use entropies based on ordinal pattern distributions as complexity measures themselves independent from the Kolmogorov-Sinai entropy, which is particularly justified by the problems mentioned and by the good performance of permutation entropy and similar measures [12,13]. For purely explorative data analysis as it is often given related to automatic learning, classification or comparison of data there is no a priori reason for avoiding large *d* and *m*.

The paper is organized as follows. We discuss the idea of ordinal patterns in Section 2, in particular, different pattern representations. Please note that the right choice of representation is substantial in developing our fast algorithms. Moreover, Section 2 provides the main entropy concept used in the paper. Section 3 is devoted to the efficient computation of ordinal patterns and ordinal pattern words from a time series. Here the special binary ordinal pattern representation mentioned above is described and used. Section 4 discusses different methods for establishing ordinal pattern and ordinal pattern word distributions. Furthermore, algorithmic complexity and runtime of our algorithms are investigated. Finally, Section 5 discusses the above specified limits of ordinal pattern-based entropies for complexity quantification in more detail. In this context, also aspects of data simulation are touched. In a certain sense, this paper is complementary to the recent paper [8] of Cuesta-Frau et. al. considering limits from the more practical viewpoint. Please consider the pseudocode descriptions of our algorithms given in the Appendix A and refer to Matlab File Exchange [14] for realization in Matlab 2018b.

## 2. Ordinal Pattern Representations and Empirical Permutation Entropy

Let $\left(X,⪯\right)$ be a totally ordered set. Mostly the set of real numbers R with the usual order ≤ is used in application. Nonetheless, our theory applies for the general case of ordinal data, for example on discrete-valued ratings or discrete rating scales as well. For our subsequent algorithmic analysis we demand that evaluation of the order relation ⪯ is not too complex and can be performed in constant time, i.e., independent from the compared elements.

We say that two vectors $x={\left(x_{k}\right)}_{k=0}^{d}\in X^{d+1}$ and $y={\left(y_{k}\right)}_{k=0}^{d}\in X^{d+1}$ share the same *ordinal pattern* of *order d* iff their components have the same order, i.e., 
$$x{∼}_{\mathrm{Ord}}y\Leftrightarrow \forall 0\leq k,l\leq d:x_{k}⪯x_{l}\Leftrightarrow y_{k}⪯y_{l}\;.$$

The equivalence relation ${∼}_{\mathrm{Ord}}$ defines (d+1)! equivalence classes on $X^{d+1}$. Please note that the definition of ${∼}_{\mathrm{Ord}}$ and the following ideas can easily be extended to the case that x and y are from different ordered sets. This expands the usage of ordinal analysis to a broader range of applications, by other means incomparable properties can be compared by these methods.

We call a single-pair-comparison within x an *atomic* ordinal information. A pair $\left(x_{k},x_{l}\right)$ with k<l is said to be *discordant* or an *inversion* if $x_{k}≻x_{l}$ and *concordant* if $x_{k}⪯x_{l}$. There are d(d−1)/2 of these elementary comparisons. Due to the transitive property of order relations, not all variations are possible to occur.

Please note that the set $\mathrm{inv}\left(x\right):=\left\{\left(k,l\right)\in \left\{0,\ldots ,d\right\}|k<l\wedge x_{k}≻x_{l}\right\}$ of all inversions within x uniquely determines the regarding ordinal pattern.

### 2.1. Ordinal Representations

Each ordinal equivalence class is represented by an ordinal pattern. We describe the ordinal patterns formally as elements of a *pattern set*
$P_{d}:=X^{d+1}/{∼}_{\mathrm{Ord}}$. Concrete representations of the ordinal patterns have to be bijections from $P_{d}$ to a suitable set. The choice of representation of the ordinal information within a pattern has an influence on various factors. These include computability, comparability and behavior under transformation as the pattern order *d* increases. The most common representations include the *permutation representation* and the *inversion vector representation*. They were considered from the beginning of Ordinal Pattern Analysis including Bandt [15] and Keller et al. [16]. Formally, they are given by the transformations
$$\begin{array}{ccccc} P: & P_{d}\to S_{d+1} & x\mapsto \pi ={\left(\pi_{k}\right)}_{k=0}^{d}, & \pi_{k} & =\#\left\{l|0\leq l\leq d\wedge \left(x_{l}≺x_{k}\vee \left(x_{l}=x_{k}\wedge l<k\right)\right)\right\},\\ I: & P_{d}\to I_{d} & x\mapsto i={\left(i_{k}\right)}_{k=1}^{d}, & i_{k} & =\#\left\{l|0\leq l<k\wedge x_{l}≻x_{k}\right\} \end{array}$$
where $S_{d+1}$ is the symmetric group on {0,1,…,d}, i.e., it contains all permutations of the numbers 0,1,…,d and where $I_{d}=\times_{k=1}^{d}\{0,\ldots ,k\}$. Please note that in the literature, the names of these representations vary; moreover some authors use the inverse permutation as the permutation representation instead.

Both transformations are based on order comparisons in different ways. The permutation representation assigns to each vector x the permutation which has the same order within its elements, i.e., for each element $x_{k}$ of x and $\pi_{k}$ of π the number of other elements in the respective vectors that are less or equal is the same. Therefore particular properties like the largest and smallest element can directly be determined from this representation. Moreover, the group structure on $S_{d+1}$ with the permutation composition operation gives many possibilities to analyze the permutation vectors in algebraic terms (e.g., Amigó [17]). However, if a new element is appended to x the whole permutation vector has to be updated. This is a main disadvantage of the permutation representation that makes computation rather inefficient.

The inversion vector representation is chosen to avoid this problem. It assigns a vector i to each x where the *k*-th entry is the number of *inversions* of $x_{k}$ with all previous elements in x. In other words, the underlying comparisons only refer to prior appended elements, therefore all preceding elements in i are not affected by the increase of *d*. This property allows a simple extension to patterns of infinite order *d*. Moreover, using the *factorial number system*, a special numeral system with mixed radix, the inversion vectors can easily be mapped to the numbers 0,…,(d+1)!−1. The map is given by $\sum_{l=1}^{d}l!\cdot i_{l}$. The resulting number representation allows the most compact description of an ordinal pattern since it combines all ordinal information in a single integer.

For these representations as well as the representation presented in Section 3 one has to note the trade-off between accessibility of the atomic information and flexibility on the one hand and compactness of the representation on the other hand.

### 2.2. Ordinal Pattern Based Entropies

In the application of the ordinal approach, experimental data $x={\left(x_{n}\right)}_{n=0}^{N}\in X^{N+1}$ of length N+1 is decoded into a sequence of ordinal patterns and, if needed, successive patterns are conflated into pattern words of length m∈N.

An important reason ordinal pattern-based measures are interesting for data analysis is that the ordinal pattern (word) sequences usually store much information about the systems dynamics if the data originates from an ergodic dynamical system. This is especially notable, if one focuses on quantifying the complexity of the underlying system. Complexity measures considered are mainly built up on the Shannon entropy of ordinal pattern (word) distributions. In addition, as already mentioned in the introduction, these measures are related to the Kolmogorov-Sinai entropy which is a central complexity measure for dynamical systems. In the following, we give precise definitions of these basic entropy concepts.

Let us regard d∈N as fixed and let p be some ordinal pattern in $P_{d}$. To determine the absolute frequency $h_{p}$ of how often p occurs in x, we identify the ordinal structures within each data segment $x^{\left(n,d\right)}:={\left(x_{k}\right)}_{k=n}^{n+d},n=0,\ldots ,N-d$. The patterns associated with the segments are denoted by $p^{\left(n,d\right)}$. Then,
$$h_{p}:=\#\left\{n\in \left\{0,\ldots ,N-d\right\}|p^{\left(n,d\right)}=p\right\}.$$

Let w be a word of m∈N successive patterns of order *d* and let $W_{d,m}$ be the set of all possible ordinal words of length *m* and order *d*. We call the parameter pair (d,m) a *word configuration*. The absolute frequency $h_{w}$ of some ordinal word $w\in W_{d,m}$, is defined by
$$h_{w}=\#\left\{n\in \left\{0,\ldots ,N-d-m+1\right\}|w^{\left(n,d,m\right)}=w\right\}$$
where
$$w^{\left(n,d,m\right)}:={\left(p^{\left(n+k,d\right)}\right)}_{k=0}^{m-1},n=0,\ldots ,N-d-m+1.$$

Please note that we write $p^{\left(n\right)}$ and $w^{\left(n,m\right)}$ when *d* is known from the context.

All entropy measures we consider in this paper, are based on quantities $H^{\left(d,m\right)};d,m\in N$. The $H^{\left(d,m\right)}$ are called *Empirical Shannon entropy of order d and word length m*, and defined by
(1)$$\begin{array}{cc} H^{\left(d,m\right)} & :=-\sum_{w\in W_{d,m}}\frac{h_{w}}{N-d-m+2}\ln\frac{h_{w}}{N-d-m+2}\\ & =\ln\left(N-d-m+2\right)-\frac{1}{N-d-m+2}\sum_{w\in W_{d,m}}h_{w}\ln h_{w}. \end{array}$$

They quantify the complexity of ordinal patterns of order *d* conflated into pattern words of length *m* in a given time series. Please note that $\frac{h_{w}}{N-d-m+2}$ is the relative frequency of such words. To compute $H^{\left(d,m\right)};d,m\in N$ one has to determinate the pattern (word) distributions, which is our main subject of the next section.

## 3. A Pattern Representation for Efficient Computation

In this section, we describe how all desired ordinal patterns in x can be computed in an efficient and fast way. We will do so by taking advantage of the redundancy in terms of atomic ordinal information between the patterns. We introduce a pattern representation that allows us to do the following tasks in a simple and fast manner when $p^{\left(n,d\right)}$ is given:Computation of the successive pattern $p^{\left(n+1,d\right)}$,Computation of patterns $p^{\left(n,d-m\right)},m=1,\ldots ,d-1$ of smaller order,Computation of the ordinal words $w^{\left(n,d-m,m+1\right)},m=1,\ldots ,d-1$.

Furthermore, our representation is chosen so that pattern and word frequencies can be determined fast.

### 3.1. Computation of Successive Patterns

Example: Consider the first six data points of a time series as shown in Figure 1 and the ordinal order d=4. Then, we have two ordinal patterns: the red colored pattern $p^{\left(0\right)}$ determined by the points $x^{\left(0\right)}={\left(x_{k}\right)}_{k=0}^{4}$ and the blue colored pattern $p^{\left(1\right)}$ determined by $x^{\left(1\right)}={\left(x_{k}\right)}_{k=1}^{5}$. Please note that both patterns are maximally overlapping and share the data points $x_{1},\ldots ,x_{4}$.

To take optimal advantage of this shared ordinal information, we subdivide the task of computing $p^{\left(1\right)}$ – when $p^{\left(0\right)}$ is given – into three parts:

**Tasklist** **1.**
*(i)* 
*remove atomic information regarding $x_{0}$,*
*(ii)* 
*adapt the atomic information regarding $x_{1}$ to $x_{4}$,*
*(iii)* 
*add atomic information regarding $x_{5}$.*



Generally, these successive steps can be realized as follows: Consider the set inv(x) of inversions within $x\in X^{d+1}$ and the lower triangular matrix $M\left(x\right)\in {\left\{0,1\right\}}^{d\times d}$
$$M\left(x\right):=\left(\begin{array}{ccc} 1_{x_{0}≻x_{1}} & & 0\\ \vdots & \ddots & \\ 1_{x_{0}≻x_{d}} & \ldots & 1_{x_{d-1}≻x_{d}}, \end{array}\right),$$
where 1 is the indicator function. It takes the value of one if the statement in the subscript is true. Otherwise, the value is zero. For instance, the schematic illustrations in Figure 2 show the overlapping atomic information between successive matrix representations in general and for our previous example.

We see that $M(x^{\left(n+1,d\right)})$ is obtained from $M(x^{\left(n,d\right)})$ in terms of Tasklist 1 by executing the following steps:

**Tasklist** **2.**
*(i)* 
*removing the first row and column of $M(x^{\left(n,d\right)})$,*
*(ii)* 
*keeping the rest of $M(x^{\left(n,d\right)})$,*
*(iii)* 
*appending a new row and column at the end with the atomic information $1_{x_{n+k}≻x_{n+d+1}}$ for all k=1,…,d.*



While the matrix representation M is suitable for computing successive patterns, it lacks of compactness and is difficult to compare. We define a more compact representation which nevertheless directly contains all atomic information – the *binary vector representation*: (2)$$B\left(x\right):=M\left(x\right)\cdot {\left(\begin{array}{ccc} 2^{0} & \cdots & 2^{d} \end{array}\right)}^{T}={\left(\sum_{0\leq l<k}1_{x_{l}≻x_{k}}2^{l}\right)}_{k=1}^{d}\in \overset{d}{\underset{k=1}{\times }}\{0,1,\ldots ,2^{k}-1\}.$$

By this definition, we encode each row of M as a binary number where the least significant bit stands in the first column of *M*. Please note that the binary vector representation is closely related to inversion vectors by
$$I\left(x\right)=M\left(x\right)\cdot {\left(\begin{array}{ccc} 1 & \cdots & 1 \end{array}\right)}^{T}.$$

By counting the ones in their binary representations, the elements of the binary vector can be transformed to inversion vector elements.

The key benefit of the binary system is that all operations on binary vectors can be performed as bit operations and can thereby be implemented in an efficient and fast way, especially in low-level programming languages. Here, we use *left* and *right bit shifts* in order to manipulate B and to use overlapping atomic information. By a right bit shift, the least significant bit gets dismissed. This corresponds to a division by 2 rounded down. Analogously, the left bit shift corresponds to a multiplication by 2 or, in other words, a concatenation of *B* with an additional 0 as the least significant bit. A bit shift on n∈N by *i* bits we denote by n≫i (resp. n≪i) for a right (resp. left) bit shift. Denote that here ≪ is a mathematical operator and should not be confused with the “much smaller than” symbol. For example, consider the number 11 which is 1011 in binary representation. Then $1011_{2}\gg 1=101_{2}$ which is 5 in decimal. Please note that bit shifts are monotonous functions.

The introduced representation simplifies Tasklist 2 to

**Tasklist** **3.**
*(i)* 
*perform a right bit shift on $B(x^{\left(n,d\right)})$ and remove the first entry,*
*(ii)* 
*keep the rest of the vector,*
*(iii)* 
*compute $\sum_{k=1}^{d}1_{x_{n+k}≻x_{n+d+1}}2^{k-1}$ and append it to the end of the vector.*



The underlying relationship between consecutive binary vectors used in Tasklist 3(i) reads as follows: Let $b^{\left(n,d\right)}={\left(b_{k}^{\left(n,d\right)}\right)}_{k=1}^{d}=B\left(x^{\left(n,d\right)}\right)$. Then(3)$$b_{k}^{\left(n,d\right)}=b_{k+1}^{\left(n-1,d\right)}\gg 1.$$

For the patterns of the example introduced in this section (see Figure 1), we have$$\begin{array}{cc} B(x^{\left(0\right)}) & ={\left(\begin{array}{cccc} 1 & 3 & 1 & 11 \end{array}\right)}^{T}\;\mathrm{and}\\ B(x^{\left(1\right)}) & ={\left(\begin{array}{cccc} 3\gg 1 & 1\gg 1 & 11\gg 1 & 4 \end{array}\right)}^{T}={\left(\begin{array}{cccc} 1 & 0 & 5 & 4 \end{array}\right)}^{T}. \end{array}$$

### 3.2. Computation of Patterns of Smaller Order

As with inversion vectors, both, the matrix and binary vector representation, share the possibility to append new elements without changing the other elements. In particular, when the pattern $p^{\left(n,d\right)}$ is already computed in terms of M, the patterns of lower order ${\left(p^{\left(n,d-k\right)}\right)}_{n=0}^{N-d}$ can be obtained by taking the (d−k)th leading principle submatrices of M as depicted in Figure 3a. In terms of B, the vector’s last *k* elements have to be removed to obtain the pattern of order d−k.

### 3.3. Computation of Ordinal Words

Ordinal words of length *m* combine the ordinal information of *m* successive patterns. Again, the scheme of atomic information shows the difference between one pattern of order *d* and two patterns of order d−1 which form a word of length 2. In the example seen in Figure 3b, the order between $x_{0}$ and $x_{5}$ is taken into account in $w^{\left(0,5,1\right)}=p^{\left(0,5\right)}$ but not in $w^{\left(0,4,2\right)}=\left(p^{\left(0,4\right)},p^{\left(1,4\right)}\right)$. Therefore, the corresponding bit $1_{x_{0}≻x_{5}}$ in $w^{\left(0,5,1\right)}$ can be set to zero in pursuance of uniting all patterns that are the same except for $1_{x_{0}≻x_{5}}$. The result corresponds with $w^{\left(0,4,2\right)}$. In the next step, $1_{x_{0}≻x_{4}}$ and $1_{x_{1}≻x_{5}}$ are set to zero to deduce $w^{\left(0,3,3\right)}$ (see Figure 3c). For general k∈{1,…,d} we have
(4)$$w_{k}^{\left(n,d-m,1+m\right)}=\left\{\begin{array}{cc} w_{k}^{\left(n,d,1\right)} & \;\mathrm{for}\;k\leq d-m,\\ (w_{k}^{\left(n,d,1\right)}\gg k-\left(d-m\right))\ll k-\left(d-m\right) & \;\mathrm{for}\;k>d-m. \end{array}\right.$$

In direct consequence of the results in Section 3.2, for ordinal words we have
(5)$$w_{k}^{\left(n,d,m+1\right)}=\left\{\begin{array}{cc} w_{k}^{\left(n,d,1\right)} & \;\mathrm{for}\;1\leq k\leq d\\ \sum_{l=1}^{d}1_{x_{k-l}≻x_{k}}2^{k-l} & \;\mathrm{for}\;d+1\leq k\leq d+m+1. \end{array}\right.$$

Since ordinal words can be treated as special ordinal patterns in our representation, we will often refer to both as patterns.

## 4. Implementation

In this part, we describe our algorithm that computes a whole series of Shannon entropies based on ordinal words of different word length and pattern order. Specifically, for a given *maximal order*
D∈N, the algorithm computes all Shannon entropies $H^{\left(d,m\right)}$ as defined in (Equation 1), where d+m≤D+1 with 1≤d,m≤D. When arranged in an array by pattern order *d* and word length *m*, this corresponds to all entries above, left, and on the main antidiagonal. These are the entropies regarding all ordinal words which contain information on at most *D* consecutive data points from x.

In Section 4.1, we will discuss different approaches and our choice. The following Section 4.2 is concerned with the algorithm’s structure. Section 4.3 is dedicated to an analysis of the complexity of our algorithm. In the end we apply our algorithm on randomly generated artificial data and analyze its speed in Section 4.4.

### 4.1. Discussion of Methods

There are several possible approaches for computing the entropy of a given data vector, which differ in terms of time and memory consumption dependent on the maximal order *D*, data length *N*. For these parameters, we have usually *N* significantly larger than *D* but smaller than D!.

To compute the empirical entropy, we need the frequencies of the occurring patterns. Here, we will discuss different ways to compute them. We demand the methods to be adaptable to the transformations (Equation 4) and (Equation 5) in a sense that the data structure should not be recomputed completely after transforming the patterns.

A first method is to iterate through all patterns and increment a frequency counter when the respective pattern occurs. It has the advantage that there is no need to store all patterns at once; instead, each pattern can be computed separately. A naïve approach is to prepare an array, which length is the number of all possible patterns, and increment its corresponding entries when a certain pattern occurs. Since the number of possible patterns increases factorially in the pattern order *D*, this method is only practical for small orders. Overall, the array would be very sparse: on the one hand, the data length *N* is far smaller than D! and thus the number of nonzero entries is relatively small. On the other hand, it was shown by Amigó [18] that the amount of forbidden patterns (i.e., patterns that cannot occur due to the inner structure of the underlying dynamics) also increases superexponentially, leading to further zero entries.

To avoid these zero entries, only the occurring patterns could be counted. This generally requires hashing for the key-value pairing that connects each pattern to its frequency. As an again simple and collision-free example, the hash can be chosen such that the patterns are indexed by their first occurrence. The disadvantage of this method is clearly that the indexing is not gained from information about the pattern itself; to achieve the hash value, it is necessary to search through all hashed patterns to find out whether the current pattern is new or did already occur. Other hash functions can be more effective, but also more complicated and not collision-free. However, it is not immediately clear how to construct them. In addition, the pattern transformations (Equation 4) and (Equation 5) would make it necessary to recompute all hashes in each step.

Since the problem is basically a searching problem, where the current pattern has to be found in the data structure for the frequencies, more advanced data structures such as search trees are promising approaches. Possible structures are B trees, AVL trees, where searching an element happens in O(D·logN) time. An extensive discussion can be found e.g., in standard text books such as the book by Cormen et al. [19]. However, search trees need an underlying total preorder on the set of ordinal patterns. Though all considered representations can be equipped with a total preorder, comparisons will typically have O(d) time complexity. This would lead to O(D·N·logN) time complexity for each entropy, or $O(D^{3}\cdot N\cdot \log N)$ for all desired entropies.

For our computations, we have chosen a different approach where all patterns are given (or computed) beforehand with a total time complexity of $O(D^{2}\cdot N\cdot \log N)$. The choice is based on our idea to compute entropies for different pattern orders and word lengths by manipulating the representation of the patterns. We explain this idea in the following subsections in detail. Specifically by considering all patterns at once our method allows computing the entropies ${\left(H^{\left(d,m\right)}\right)}_{m=1}^{D+1-d}$ in one run, which reduces the complexity magnitude of *D* by one. Though our methods result in the same asymptotic complexity for a single entropy, our experiments show that our approach is faster due to the possibility to vectorize the operations.

### 4.2. Structure of the Algorithm

We present the design of our algorithm in different ways. First we give a short overview of its structure with schematic illustrations. In the following four parts of the section, we explain key ideas of the algorithm in detail. A pseudocode implementation is provided in Algorithm A1 in the Appendix A. An implementation for Matlab 2018b is provided by the authors on Matlab File Exchange [14].

In Figure 4, the algorithm is depicted as a workflow diagram. The basic idea is to precompute all the words $w^{\left(\cdot ,D,1\right)}$ and use them to compute all other word configurations by (Equation 4) and (Equation 5), i.e., $w^{\left(\cdot ,d,m\right)}$ with d≤D and m≤D−d+1. For each word configuration we then compute the respective entropy. We start with $w^{\left(\cdot ,D,1\right)}=p^{\left(\cdot ,D\right)}$ since this is the word which contains all atomic information about the D+1 considered points. As shown in the preceding section, other word configurations can be obtained by successively removing the atomic information through bit manipulations.

#### 4.2.1. Initial Pattern Computation

The initial computation of the $p^{\left(\cdot ,D\right)}$ is shown schematically in Figure 5. The corresponding pseudocode implementation can be found in Algorithm A2 in the Appendix A. It follows the idea of Tasklist 3, but operates in a different sequence. Here, at first the last entry $p_{D}^{\left(\cdot ,D\right)}$ is computed for all patterns. The penultimate elements $p_{D-1}^{\left(\cdot ,D\right)}$ are computed by performing a right bit shift on these last entries (using k=D−1 in (Equation 3)). Sequentially all prior elements $p_{D-2}^{\left(\cdot ,D\right)},\ldots ,p_{2}^{\left(\cdot ,D\right)},p_{1}^{\left(\cdot ,D\right)}$ get computed in this way. In comparison to computing all patterns consecutively, the computation steps of this methods are highly parallelizable and vectorizable and thereby faster to perform.

#### 4.2.2. Obtaining other Word Configurations

The way our main algorithm processes through the different word configurations after computing the $p^{\left(\cdot ,D\right)}=w^{\left(\cdot ,D,1\right)}$ is illustrated in Figure 6. In the main iteration using (Equation 4) the pattern order gets successively decreased while the word length gets increased. For a fixed pattern order *d* and word length *m*, all words with length smaller than *m* can be computed by simply cutting of last elements of the binary vectors as (Equation 5) indicates.

#### 4.2.3. Computation of the Pattern Frequencies

With each list of words, we compute the word frequencies. In short, the frequencies are determined by sorting the patterns and by computing the length of blocks of the same pattern. Thus, as the first step, all words are sorted lexicographically. Next, we compute the elementwise differences
$$w^{\left(n,d,m\right)}-w^{\left(n-1,d,m\right)}={\left(\begin{array}{cccc} w_{1}^{\left(n,d,m\right)}-w_{1}^{\left(n-1,d,m\right)} & w_{2}^{\left(n,d,m\right)}-w_{2}^{\left(n-1,d,m\right)} & \ldots & w_{d}^{\left(n,d,m\right)}-w_{d}^{\left(n-1,d,m\right)} \end{array}\right)}^{T}$$
between successive patterns. Since the patterns are sorted, the difference vectors are zero vectors for identical patterns and have positive entries for different patterns. The signs of the cumulated differences within the patterns, given by
$${\left(\mathrm{sgn}\left(\sum_{l=1}^{k}\left(w_{l}^{\left(n,d,m\right)}-w_{l}^{\left(n-1,d,m\right)}\right)\right)\right)}_{k=1}^{d}$$
indicate a pattern change for all word lengths 1,…,d. The cumulation reflects the property that—due to the lexicographic order—changes in prior entries imply different patterns for all higher pattern lengths. As the positive signs indicate a change of patterns, the pattern frequencies for the words of length *m* can be determined by taking the difference between the indices of patterns that have positive signs in the *m*-th element. With the computed absolute frequencies $h_{p}$, the corresponding entropy can be computed with (Equation 1). Refer to Algorithm A3 in the Appendix A for a pseudocode implementation.

#### 4.2.4. Inclusion of Missing Lower Order Patterns—The Frequency Trick

Not all ordinal words can be computed by transforming the $p^{\left(\cdot ,D\right)}$ into other word configurations. This is due to the fact that applying (Equation 4) and (Equation 5) does not change the total number of patterns although the number of patterns contained in the data increases for other configurations. With a total data length N+1, we get N+1−D patterns ${\left(p^{\left(n,D\right)}\right)}_{n=0}^{N-D}$ in the beginning. For arbitrary pattern order *d* and word length *m*, we get N−d−m+2 words ${\left(w^{\left(n,d,m\right)}\right)}_{n=0}^{N-d-m+1}$. Since d+m−1≤D holds by assumption, we have N−d−m+1≥N−D. Thus, the last D−d−m+1 patterns cannot be derived from the $p^{\left(\cdot ,D\right)}$. We illustrate the missing patterns in Figure 7.

The procedure would become complicated if all the missing different sized patterns are computed and stored separately. Instead, we desire to compute all patterns by our methods. We achieve this by expanding the data vector with artificial data such that now the missing patterns of the enlarged data vector contain only artificial data and can be neglected. The word $w^{\left(N-1,1,1\right)}$ is the last occurring ordinal word in x of all configurations; it contains the ordinal information between the last points $x_{N-1}$ and $x_{N}$. By going backwards through the computation process, it can be seen that $w^{\left(N-1,1,1\right)}$ is obtained from $w^{\left(N-1,D,1\right)}$. This pattern is determined by the points $x^{\left(N-1\right)}={\left(x_{k}\right)}_{k=N-1}^{N+D}$ which are yet undefined for N+1≤k≤N+D. Therefore it suffices to pad these N+D−(N+1)=D−1 elements to x. With the padded data, we get *N* patterns in total, which is sufficient for each word configuration.

For convenience, we define an element ∞∉X to be the unique value such that ∞≻y for all y∈X. By padding x with *∞*, each atomic information about artificial elements as well as the regarding bit in the binary vector representation is always set to zero.

When the patterns are computed based on the padded data, each required word can be deduced from the *N* initially computed patterns. As stated above, we have to consider only the first N−d−m+2 words ${\left(w^{\left(n,d,m\right)}\right)}_{n=0}^{N-d-m+1}$ for the entropies. The last d−m+2 words ${\left(w^{\left(n,d,m\right)}\right)}_{n=N-d-m+2}^{N-1}$ partly contain ordinal information regarding the artificial data and have to be excluded from the entropy computations. Nonetheless, the latter are still needed for deducing words with other configurations and, thus, they cannot simply be deleted.

To prevent these words from distorting the word frequencies without removing them, we use a property of the entropy formula that we call the *frequency trick*. When the entropy is written in terms of absolute frequencies as in (Equation 1), patterns of absolute frequency 1 have no effect to the summation due to ln1=0. We take advantage of this by adjusting the binary vectors such that
(6)$$h_{w^{\left(n,d,m\right)}}=1\;\mathrm{for}\;N-d-m+2\leq n\leq N-1.$$

Thus, we can keep these patterns in our computing procedure. We can use the entropy Formula (Equation 1), although the number of pattern $\sum_{w\in W_{d,m}}h_{w}=N-1$ is higher than the expected number of patterns N−d−m+2.

We can make all words deduced from the initial patterns satisfy (Equation 6) only by adjusting the additional patterns ${\left(p^{\left(n,D\right)}\right)}_{n=N-D+1}^{N-1}$. This can be done by replacing the artificial zero entries as follows: $$\begin{array}{ccccccccc} p^{\left(N-D+1\right)} & = & (p_{1}^{\left(N-D+1\right)} & p_{2}^{\left(N-D+1\right)} & p_{3}^{\left(N-D+1\right)} & \ldots & p_{d-2}^{\left(N-D+1\right)} & p_{d-1}^{\left(N-D+1\right)} & 2^{D}),\\ p^{\left(N-D+2\right)} & = & (p_{1}^{\left(N-D+1\right)} & p_{2}^{\left(N-D+2\right)} & p_{3}^{\left(N-D+2\right)} & \ldots & p_{d-2}^{\left(N-D+2\right)} & 2^{D-1} & 2^{D}),\\ & \vdots & & & & & & & \\ p^{\left(N-2\right)} & = & (p_{1}^{\left(N-2\right)} & p_{2}^{\left(N-2\right)} & 2^{3} & \ldots & 2^{D-2} & 2^{D-1} & 2^{D}),\\ p^{\left(N-1\right)} & = & (p_{1}^{\left(N-1\right)} & 2^{2} & 2^{3} & \ldots & 2^{D-2} & 2^{D-1} & 2^{D}). \end{array}$$

The values are chosen in a way that they are larger than the greatest possible value for each entry. Indeed, in the binary vector representation the *k*-th entry is an integer based on *k* bits, which reaches its maximum with all bits being ones, which is $\sum_{0\leq l<k}2^{l}=2^{k}-1$. This choice makes it possible to identify any of the additional patterns independent from the data. Thus, each of the additional patterns is fully unique.

On top, each word that is deduced from one of the additional patterns is unique, as long as it contains ordinal information about artificial data points. Recall that the entries with artificial information has values larger than all entries with real information at the same index. Since bit shifts are monotonously increasing functions, applying (Equation 4) to each entry keeps the values of the artificial entries to be larger. Furthermore, by (Equation 5), only the last elements are removed. After applying both transformations, $p^{\left(N-D+1\right)}$ contains no more artificial ordinal information. This is consistent with the increase of the number of patterns when *m* decreases. The other patterns $p^{\left(N-D+2\right)},\ldots ,p^{\left(N-1\right)}$ stay unique. In summary, both transformations keep the uniqueness of the patterns as long as artificial information remains and (Equation 6) is satisfied.

### 4.3. Complexity Analysis

In the following, we analyze our algorithms both theoretically and practically. First we give an analysis in terms of Landau Big-O notation dependent on the parameters *N* and *D*. It is summarized in Figure 8. Take into account that in general it is not trivial to extend Big-O notation to the multidimensional case (c. f. [20]). Nonetheless, since D≪N holds, the asymptotics of our algorithm is mainly determined by *N*. Therefore, the one dimensional definitions of Big-O notation apply. Further note that in terms of complexity, the base of considered logarithms has no influence. Therefore we denote the logarithm with the generic log in this section.

We analyze the algorithm on the individual parts as divided in Figure 4. In terms of time complexity, the initial computation of the patterns discussed in Section 4.2.1 needs N−D+1 comparisons for the last pattern row. For each of the D−2 following rows, one additional comparison and N−D−1 bit shifts are performed, which leads to $DN-D^{2}+2D-2N+2$ bit shifts and N−1 comparisons. Assuming both operations can be performed in constant time, we get a time complexity of $O(D^{2}+D\cdot N)$. Since usually *N* is significantly larger that *D*, it can be simplified to O(D·N). In comparison, the naïve approach of finding all ordinal patterns in the binary vector representation by performing all D2 comparisons for each pattern has a worse total time complexity of $O(D^{2}\cdot N)$.

For computing the other representations presented in Section 2 patternwise, there exist O(DlogD·N) algorithms; the permutation representation can be obtained by sorting the data, for the inversion vector representation Knuth [21] gave an O(DlogD) conversion from permutations. It is known that DlogD is a lower boundary for sorting algorithms based on comparisons [19]. Thus in terms of complexity, these algorithms are optimal for these representations when computing each pattern by itself. Adapting information from previous patterns allowed us to achieve the improvement in terms of linear time complexity in *D*. In similar ways, there are ways to obtain linear time complexity for permutations and inversion vectors. The latter was shown by Keller et al. [16], their idea can be adapted for the former straightforward.

The padding of the artificial data can be realized in O(D) insertions which amortized needs O(D) time. the data is stored in a dynamic array. The adjustment of the last patterns concerns $\frac{1}{2}D\left(D-1\right)\in O\left(D^{2}\right)$ elements. In total, the precomputation of the patterns happens in O(D·N) time.

In the next step, the patterns get sorted. For this, typical sorting algorithms can be used. Pattern comparisons need *D* comparisons of their elements in the worst case. Hence the sorting needs O(D·NlogN). Please note that in practice, there are less comparisons needed. The probabilities of the first *k* elements being equal is quickly decreasing in *k*; in particular, it is given by $\sum_{w\in W_{d,m}}\mathrm{Pr}{\left(w\right)}^{2}$, where Pr denotes the (usually) unknown probability of the word w.

The inner loop takes O(N) time: The transformation takes no additional time since it can be reached by simply ignoring the respective last elements. To attain the changes between the sorted patterns and thereby the frequencies, an iteration through all patterns is needed; for the entropy, all the O(N) pattern frequencies must be processed. All iterations need O(D·N) time altogether. Each of the subsequent transformations of the patterns consists of *D* bit shifts. Together with the resorting of the transformed patterns, all O(D) iterations take $O(D^{2}\cdot N\log N)$ time in total.

In summary, our algorithm has a time complexity of $O(D^{2}\cdot N\log N)$ for $\frac{1}{2}D\left(D+1\right)$ entropies. In principle, for a single entropy each of the loops has to be performed only once, leading to the complexity of O(D·NlogN).

### 4.4. Runtime Analysis

In this section, we test our algorithm on artificial data in order to evaluate its performance and compare it to other available programs. The artificial data is generated by a simple random number generator which gives uniformly distributed random numbers on [0,1]. The data is chosen to be uniformly distributed because under this assumption also the ordinal patterns are uniformly distributed. This corresponds to the Shannon entropy taking its maximum. In this sense, the case of uniformly distributed data can be considered as the worst case scenario from the computational viewpoint. In contemplation of investigating the independence of the results from the distribution, we additionally performed our algorithms on normally distributed data. Our computations were performed on Matlab 2018b on a Windows 10 computer equipped with an Intel Core i7-3770 CPU and 16 GB RAM. We repeated the computations fifty times in order to exclude external distortions and took the mean value of the results.

#### 4.4.1. Computational Time for Pattern and Entropy Computation

First, we restrict our analysis to the pattern computation. We compare our binary vector approach with implementations which determines the inversion and permutation vector representations. For all three representations, we compare the naïve, patternwise approach with the successive methods introduced in Section 3.

As shown in the prior section, the computational time depends on the data length *N* and the maximal pattern order *D*. In our implementation in Matlab, both parameters are limited due to the following reasons. In a binary vector of order *D*, its elements are integers with up to *D* bits. Since arithmetic computations are needed when the bit shifts are performed, *D* is physically limited to 64 when using unsigned long integers. In practice, it is even limited to 51 since internally, all arithmetic calculations are performed on floating-point numbers in Matlab. From the 64 bits only 52 bits are reserved for the mantissa from which one bit is needed for the frequency trick. Recall that the maximal possible precision in Matlab is given by approximately $2.2204\cdot 10^{-16}$.

The data length is in our implementation limited by the machine’s memory. In our case, a 16 GB RAM leads to approximately $4\cdot 10^{9}$ array elements. Therefore, to store all *N* patterns of order *D*, $N\approx 10^{8}$ data points are possible to process on our machine.

Clearly, for Matlab or other programming languages there are several possibilities for memory management and high-precision integer arithmetics (c. f.  the vpi toolbox [22]), which are offered either by the maintainer or by third-party programs. Anyhow, we restricted ourselves to basic methods of Matlab.

We tested the runtime of our algorithm for varying *N* and *D* on randomly generated data vectors by usage of Matlabs tic and toc commands, which measure the time between both function calls. The results can be seen in Figure 9a–d. We compare uniformly distributed Figure 9a,b and normal distributed Figure 9c,d data. Both distributions lead to similar results. For both, the asymptotic tendency found in the previous section is observable in the graphs: All implementations have linear tendency in *N*. The successive implementations are also linear in *D*, whereas a superlinear behavior can be examined on the naïve approaches. For each representation, the successive implementation is significantly higher than its naïve counterpart. In addition, the successive binary vector approach is the fastest in total. On average, it is 5.38 times faster than the successive algorithm for the inversion vectors and 5.89 times faster than the respective permutation vector algorithm.

In addition to the runtime tests for the patterns, we tested the time consumption of the whole program. Again, the theoretical considerations of Section 4.3 are confirmed. For increasing data length *N*, the runtime follows an N·logN curve. We assume that the deviation for small *N* is due to the strictly linear time complexity of pattern computation. The proportion of computing time for the patterns tends to 10%. For varying order *D*, the quadratic regression we did on our results for the parameter *D* gives an almost perfect fit (compare Figure 9f). With increasing maximal pattern order *D*, the proportion of computing time for the pattern calculation relative to the total time decreases considerably. Most of the computation time is spent on the frequency computation and the sorting of the patterns. Therefore, this part seems to be a promising starting point for future improvements.

#### 4.4.2. Comparison with Other Implementations

We compare our algorithm to the implementations by V. Unakafova (see *PE.m*, [23], detailed explanations in the accompanying literature [24]), G. Ouyang (*pec.m*, [25]) and A. Müller (*petropy.m* [26], description in [9]) in the context in which the programs are comparable. None of the other implementations can compute entropies for ordinal words. The program *pec.m* computes a single entropy for a given pattern order and supports time delays. *petropy.m* again gives only one entropy, but allows multiple time delays. In addition it offers several ways to treat equal values in the data. The implementation in *PE.m* computes the permutation entropy for sliding windows over the data set, giving the entropy for each window.

Therefore we restrict our comparisons to the case of simple ordinal patterns (m=1). Though our algorithms can be extended easily to support time delays, we will not use time delays in any implementation. At last the sliding window in *PE.m* was chosen such that the number of windows equals one.

For further comparability, we divide the runtime for our algorithm by the number of computed entropies $\frac{1}{2}D\cdot \left(D+1\right)$ to get an approximation for the mean time needed for a single entropy. In addition, we tested a modified version of our algorithm that computes the entropy for a single word configuration.

Figure 10 shows the results of our analysis. Again, we analyzed the runtime in dependence of the data length *N* on the left and pattern order *D* on the right.

In *pec.m*, the first method from our discussion in Section 4.1 was chosen. This led to very long computing times even for small *N* and *D*, making it by far the slowest approach (the green line in Figure 10).

The performance of *PE.m* (purple line) varies. It is clearly the fastest of the compared approaches for small pattern orders. Though, for d=8, bigger data sets are needed to compete with the theoretical average time per entropy of our method. Responsible for the high computation speed is mainly the usage of lookup tables to determine succeeding patterns. While this method clearly reduces computation time, these tables have to be available and increase superexponentially in size, which limits its usage to pattern orders of size 8 or smaller. With increasing size of the lookup table, its speed advantages decrease.

The implementation of *petropy.m* (cyan line) has a similar performance to our modified one-entropy-programm, mainly because the program uses a similar approach to ours. It uses the same method of frequency determination. However, it uses the factorial number representation for the ordinal patterns. This choice leads to the upper limit of the pattern order since these numbers can be as high as D!. With an maximal possible integer value of $2^{53}$ in Matlab, this results in D=13 as the largest possible pattern order, a slightly higher range for *D* than *PE.m*.

In direct comparison, our approach that computes all $\frac{D\cdot (D+1)}{2}$ entropies takes (apart of the by far slowest *pec.m*) the most time for computing. On the other hand, the average time per entropy is on wide parameter ranges the fastest. Only for small *d*, the approach of V. Unakova is faster. The modified version competes well with the other approaches. Nevertheless it is slower than the average computing time per entropy since the structures used for optimizing the computation of multiple entropies cannot be exploited.

Clearly, the main advantage of our algorithm lies in the extended parameter range for the pattern order *d*. If over the top the behavior of the entropy in dependence of *d* is from particular interest, our approach saves much time compared to computing alls the entropies serially.

## 5. Limits of Ordinal Pattern Based Entropies

As already mentioned in the introduction, most complexity measures for dynamical systems are strongly linked to the Kolmogorov-Sinai entropy (abbrev KSE). In our following discussion of ordinal pattern-based entropies in that context, we forgo to give detailed definitions of this concept. Instead, we refer for details and more background of the following to [13]. Furthermore we restrict the following considerations to the most simple case of a one-dimensional dynamical system. Please note that with the appropriate generalizations, stochastic processes could be included in the discussion withal.

By a (measurable) *dynamical system*
(X,A,T,μ) we understand a probability space (X,A,μ) equipped with a map T:X↩ being measurable with respect to A, where μ is invariant with respect to *T*. The latter means that $\mu (T^{-1}\left(A\right))=\mu \left(A\right)$ for all events A∈A and describes stationarity of the system. *X* is considered as the *state space* and *T* provides the dynamics. A dynamical system as given produces a time series $x={\left(x_{n}\right)}_{n=0}^{N}$, when $x_{0}\in X$ is given and $x_{n}$ is the *n*-th iterate of $x_{0}$ with respect to *T* for n=1,2,…,N. If the probability measure is ergodic, as we assume in the following, statistical properties of the given system can be assessed from such time series for sufficiently large *N*. In particular, probabilities of patterns can be estimated by their relative frequencies within the time series. With this setting there are several ways for estimating the Kolmogorov-Sinai entropy from ordinal pattern-based entropies given a time series from the system.

In theory, the following quantifiers had shown to be good estimators of the Kolmogorov-Sinai entropy for sufficiently large N,d and *m*:(7)$$\begin{array}{c} \frac{1}{d}H^{\left(d,1\right)}\mathrm{if}\;T\mathrm{is}\mathrm{an}\mathrm{interval}\mathrm{map}\mathrm{with}\mathrm{countably}\mathrm{many}\mathrm{monotone}\mathrm{pieces}, \end{array}$$(8)$$\begin{array}{c} \frac{1}{m}H^{\left(d,m\right)}\mathrm{and}H^{\left(d,m+1\right)}-H^{\left(d,m\right)}\mathrm{generally}. \end{array}$$

Please note that $\frac{1}{d}H^{\left(d,1\right)}$ is called the *empirical Permutation entropy of order d* (abbrev. ePE) and $H^{\left(d,2\right)}-H^{\left(d,1\right)}$ the *Empirical Conditional entropy of Ordinal Patterns of order d* (abbrev. eCE). Both are estimators of the respective entropies of the dynamical system which base on the time series x and whose precise definitions are specified in the given literature.

For (Equation 7) the statement follows directly from a recent result of Gutjahr and Keller [10] generalizing the celebrated result of Bandt et al. [3] that for piecewise monotonous interval maps the KSE and the Permutation entropy are coinciding. As reported in [12,13], the latter entropy seems to be a good estimator of the KSE for sufficiently large *d* with some plausible reasoning, but this point is not completely understood.

The problem in the statement above is what sufficiently large means. First of all, in the case of existence of many different ordinal patterns in a system, *N* must be extremely large in order to realize them all in a time series (for practical recommendations relating *N* and *d*, see e.g., [8,9]). Even if arbitrarily long time series would be possible, there would be serious problems to reach the KSE. Let us demonstrate this for the ePE (compare (Equation 7)).

For d∈N it holds
(9)$$\frac{1}{d}H^{\left(d,1\right)}\leq \frac{\ln((d+1)!)}{d}=\frac{\sum_{k=1}^{d+1}\ln k}{d}$$
since maximal Shannon entropy is given for the uniform distribution. The right side of this formula is very slowly increasing. For example, for d=100 it is less than 4, saying that for maps with KSE larger than 4 one needs ordinal patterns of order larger than 140 to be able to get a sufficiently good estimation. The larger the KSE is, the larger the *d* for its reliable estimation theoretically must be, and the KSE can be arbitrarily large. One reason for this is that if a map *T* has some KSE *c*, then its *n*-th iterate $T^{\circ n}$ has KSE n·c (see e.g., [27]).

Formula (Equation 9) is also interesting from the simulation viewpoint. Given an (one-dimensional) dynamical system and some precision, then usually simulations stick to the precision: Given a value in its precision, the further iterates in their precision are determined. This shows that the precision bounds the number of possible ordinal patterns independent of *d*.

For example, consider the state space [0,1] and the usage of double-precision IEEE 754 floating-point numbers, which is the standard data type for numerical data in the most programming languages. Recall that these numbers have the form $s\cdot m\cdot 2^{e}$, where s,m and *e* are coded within 64 bits of information. One bit is reserved for the sign *s* and 11 bits are used for the exponent *e*. The remaining 52 bits are left for the fraction value (mantissa) *m*. The exponent generally ranges from −1022 to 1023, for our particular state space, only negative exponents are needed. For each choice for the exponent, there are $2^{52}$ possible values of the mantissa. Therefore, there are in total $1022\cdot 2^{52}\approx 4.6\cdot 10^{18}$ distinct numbers possible in a simulation, and more distinct ordinal patterns cannot be found. Since already for d=20 there are $21!\approx 5.1\cdot 10^{19}>4.6\cdot 10^{18}$ possible ordinal patterns, simulations do not provide an ePE larger than $\frac{\ln21!}{d}\leq \frac{\ln21!}{20}\leq 2.269$. Because there are one-dimensional maps with arbitrary KSE, this limits the estimation of the KSE by the simulation.

The kind of simulation described, which we refer to as the naïve simulation, does not reproduce the real-world situation. The measuring process itself goes hand in hand with precision loss and therefore measuring errors occur. By iterating through these erroneous values, the error cumulates. In consequence due to the chaotic behavior, the values obtained after a few iterations have nothing left in common with the real values. For example, when identifying a real-world system with a dynamical model, the simulation can in consequence produce periodic sequences of values, although the sequence of measured values does not contain periodicities. This can reduce the number of ordinal patterns and so corresponding entropies.

In real-world systems, the measuring process and its belonging errors have no influence to the system. There—under assumption of other sources of noise—the system generates its time-dependent values of arbitrary accuracy exactly. Only in the moment where we want to know a certain iterate, we measure it with the given limited precision. Therefore, we desire to have a simulation where the iteration itself can be performed exactly for starting values of unlimited precision.

Here for demonstration purposes we are especially interested in the system ([0,1],B,L,μ), where B are the Borel sets on [0,1], *L* is the logistic map defined by L(x)=4x(1−x) for x∈[0,1] and μ is the probability measure on B with a density *p* on [0,1] given by $p\left(x\right)=\frac{1}{\pi \sqrt{x(1-x)}}$ for x∈]0,1[.

To mimic the real situation, we generate a sequence $x_{0}^{\prime },x_{1}^{\prime },x_{2}^{\prime },\ldots$ by using other systems that are topologically (semi-)conjugated to the logistic map. These systems show to be suitable to our simulation requirements.

In the first step, we take advantage of the well-known fact that the system ([0,1],B,L,μ) is semi-conjugated to the angle doubling map on [0,2π] which is given by A(β)=2βmod2π equipped with the Lebesgue measure (compare [28] for this and the following statements). The semi-conjugacy is given by the map ψ with
$$\psi \left(\beta \right)=\sin^{2}\left(\beta \right)$$
for β∈[0,2π]. This means that
(10)L(ψ(β))=ψ(A(β))
for all β∈[0,2π], i.e., applying *L* on [0,1] corresponds to doubling the arc length of a segment on the unit cycle and thereby doubling the angle. Furthermore, μ is the image of λ, i.e., $\mu \left(B\right)=\lambda (\phi^{-1}\left(B\right))$ for all B∈B. The relationship is illustrated in Figure 11.

As the second step, we use another conjugacy, namely that the angle doubling map is conjugated to the shift function on all infinite 0-1-sequences, i.e., ${\left\{0,1\right\}}^{\infty }$ equipped with the $\left(\frac{1}{2},\frac{1}{2}\right)$-Bernoulli measure ν. The latter assigns to each cylinder set of size *n* of ${\left\{0,1\right\}}^{\infty }$ the measure $2^{-n}$. The shift is defined as
$$S:{\left\{0,1\right\}}^{\infty }↩\;,\left(b_{1},b_{2},b_{3},\ldots \right)\mapsto \left(b_{2},b_{3},b_{4},\ldots \right).$$

The conjugation between a 0-1-sequence ${\left(b_{j}\right)}_{j\in N}$ and an angle β∈[0,2π] is given by the binary expansion
$$\beta =\phi \left({\left(b_{j}\right)}_{j\in N}\right)=2\pi \sum_{j=1}^{\infty }b_{j}2^{-j}.$$

Please note that ϕ is only almost everywhere bijective, for example ϕ((1,0,0,…))=ϕ((0,1,1,…))=π. Analogously to (Equation 10), it follows that
(11)$$A\left(\phi \left({\left(b_{j}\right)}_{j\in N}\right)\right)=\phi \left(S\left({\left(b_{j}\right)}_{j\in N}\right)\right)$$
for all ${\left(b_{j}\right)}_{j\in N}\in {\left\{0,1\right\}}^{\infty }$. Concluding (Equation 10) and (Equation 11), the following diagram commutes:$$\begin{array}{ccccc} \left(\{0,1\}{}^{\infty },\nu \right) & \overset{\phi }{\to } & \left([0,1],\lambda \right) & \overset{\psi }{\to } & \left([0,1],µ\right)\\ ↓S & & ↓A & & ↓L\\ \left(\{0,1\}{}^{\infty },\nu \right) & \overset{\phi }{\to } & \left([0,1],\lambda \right) & \overset{\psi }{\to } & \left([0,1],µ\right) \end{array}$$

We can generate an orbit $x_{0},x_{1},x_{2},\ldots$ for *L* as follows: Take a random sequence $b_{1},b_{2},b_{3},\ldots ,$ of elements in {0,1} meaning that each $b_{n}$ is taken from the symbols {0,1} with equal probability and in an independent way and let
$$x_{n}=\psi (\phi \left(S^{\circ n}\left({\left(b_{j}\right)}_{j\in N)}\right)\right)=\sin^{2}\left(2\pi \sum_{j=1}^{\infty }b_{j+n}2^{-j}\right).$$

Let p∈N be the number of binary digits defined by a given precision. The approximation up to precision *p* given by
$$x_{n}^{\prime }=\sin^{2}\left(2\pi \sum_{j=1}^{p}b_{j+n}2^{-j}\right).$$
provides a sequence $x_{0}^{\prime },x_{1}^{\prime },x_{2}^{\prime },\ldots$ as desired. The pleasant point is that in the simulation we can start from $b_{1},b_{2},\ldots ,b_{p}$ and than can append $b_{p+1}$ by random choice and delete $b_{1}$, then add $b_{p+2}$ and delete $b_{2}$, then add $b_{p+3}$ and delete $b_{3}$, etc. We refer to this simulation (see Algorithm 1) as the advanced one.


**Algorithm 1:**
Logistic-Map-Simulation


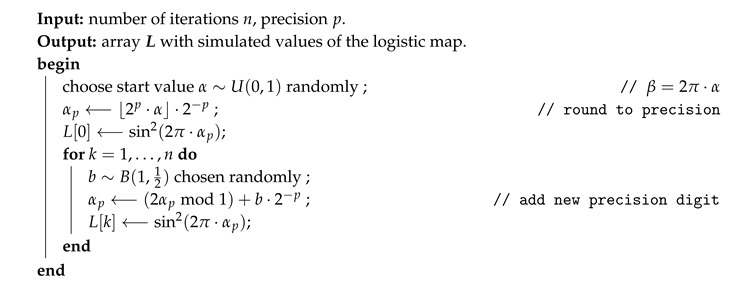



We have determined the ePE and the eCE for time series related to the logistic map *L* and its iterates $L^{\circ 2}=L\circ L,L^{\circ 3}=L\circ L\circ L$ in dependence on the order *d*. The results based on time series of length $10^{7}$ are presented by Figure 12. The time series behind the pictures on the left side ((a) and (c)) was obtained by the naïve simulation and on the right side ((b) and (d)) by the advanced simulation, both under the double precision described above. In (a) and (b), we have added **blue** colored curves showing the upper bound of the ePE (Equation 9) for *d*. Please note that the results were obtained by our algorithm from Section 4. It allowed to compute all values at once for each curve.

First of all, for *L* and $L^{\circ 2}$ (**red** and **yellow** colored curves) with KSE ln2≈0.693147 and 2ln2≈1.38629, respectively, the values of eCE restore the KSE much better than the ePE. For a very good performance of the eCE, consider *d* in the interval between 6 and 16 for *L* and between 6 and 9 for $L^{\circ 2}$. For *d* left of the intervals, the ordinal patterns seem to be too short to capture the full complexity, for *d* right of the intervals the time series seem to be to short relative to *d* to capture the ordinal pattern complexity. As already reported in [12,13] for *L*, convergence of the ePE seems to be too slow to get good estimations for moderately long time series.

It is not excluded that for *L* or $L^{\circ 2}$ the naïve algorithm sometimes runs in a loop or, more general, reduces variety of ordinal patterns. For $L^{\circ 3}$, we see such behavior in Figure 12 (cf. the **purple** curves), namely larger empirical entropies are reached for the advanced simulation than for the naïve one. It is remarkable that the eCE for d=8 is near to the KSE 3ln2≈2.07944, it seems however that the set of too small and the set of too large *d* as described above are overlapping, such that the KSE is not reached completely.

Our exemplary considerations underline that some deeper thinking about the (mathematical) structure of systems and measurements, possibly also on the base of symbolic dynamics, is important for simulating data and beyond. To develop powerful and reliable tools for data analysis, moreover, a better understanding of ordinal pattern-based entropies for complexity quantification and their relationship to other complexity measures remains a permanent challenge.

## Figures and Tables

**Figure 1 entropy-21-00547-f001:**
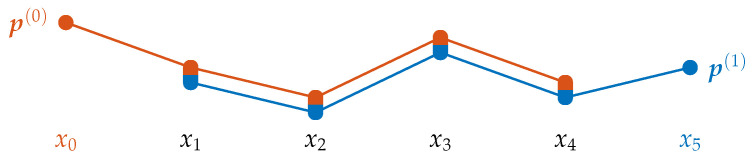
Successive ordinal patterns of order d=4.

**Figure 2 entropy-21-00547-f002:**
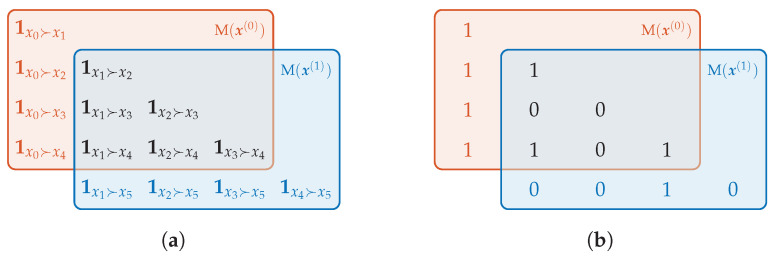
Schematic representation of the overlap between successive matrix representations $M(x^{\left(0\right)})$ and $M(x^{\left(1\right)})$ in general (**a**) and for our example (**b**) (compare to Figure 1).

**Figure 3 entropy-21-00547-f003:**
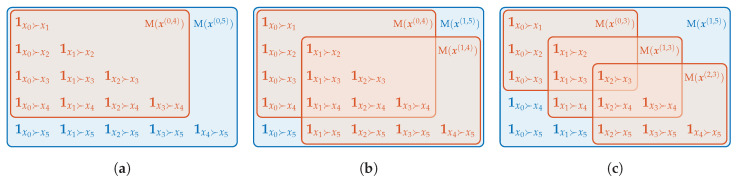
(**a**) Relationship between patterns of different order. (**b**) Difference between a word of length 2 and order 4 and the pattern of order 5. (**c**) Difference between a word of length 3 and order 3 and the pattern of order 5.

**Figure 4 entropy-21-00547-f004:**
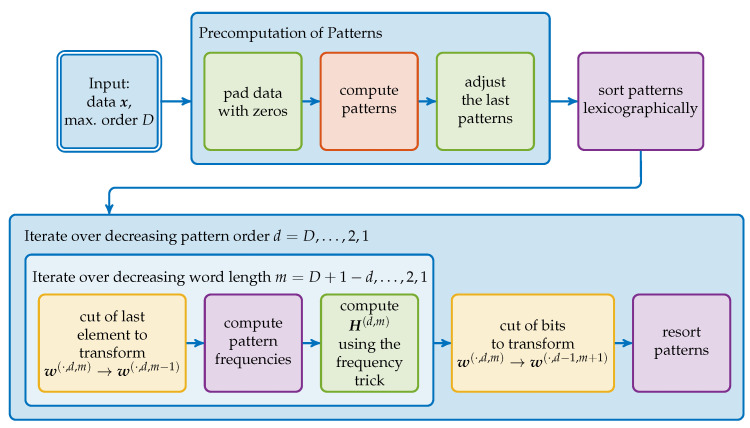
Schematic overview ofr our algorithm. The colors of the steps refer to different aspects of the algorithm discussed in the following subsections: The initial pattern computation (**red**), the processing of these patterns to other word configurations (**yellow**), the calculation of the pattern frequencies (**purple**) and the modifications to include all lower order patterns (**green**).

**Figure 5 entropy-21-00547-f005:**
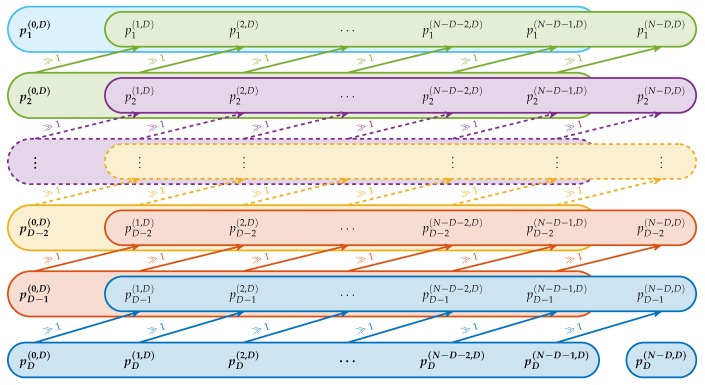
Computation scheme for the ordinal patterns, where each column corresponds to a binary vector. Each color shows one iteration step. The **bold** elements are computed directly while all other elements are derived from them by bit shifts. These are indicated by the arrows.

**Figure 6 entropy-21-00547-f006:**
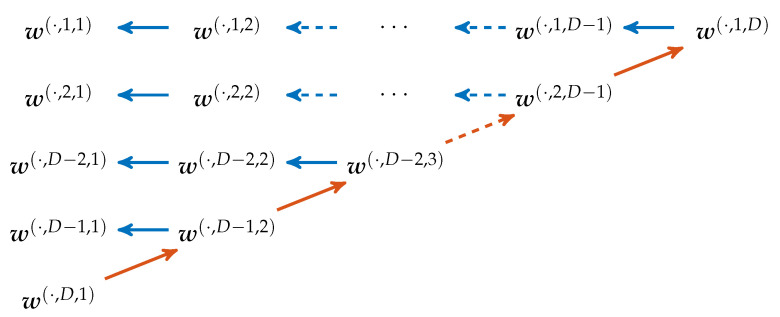
Computation of the different words in the algorithm. **Red** arrows denote applications of (Equation 4), while **blue** arrows mean usage of (Equation 5).

**Figure 7 entropy-21-00547-f007:**
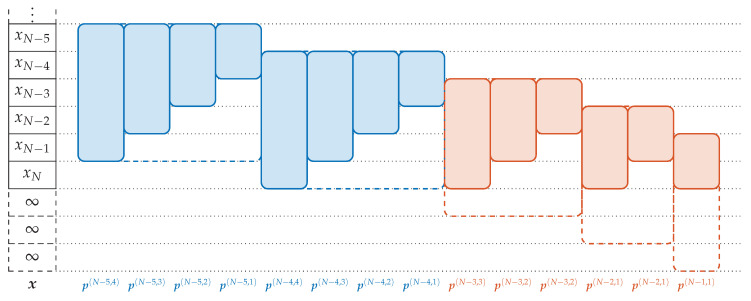
The patterns which are not covered by $p^{\left(\cdot ,D\right)}$ for D=4 and m=1. The **blue** patterns are the two last patterns of $p^{\left(\cdot ,D\right)}$ and the derived patterns of smaller order. The **red** patterns are the missing patterns. They can be derived from the dashed patterns constructed by padding D−1=3 elements with value *∞* at the end of x.

**Figure 8 entropy-21-00547-f008:**
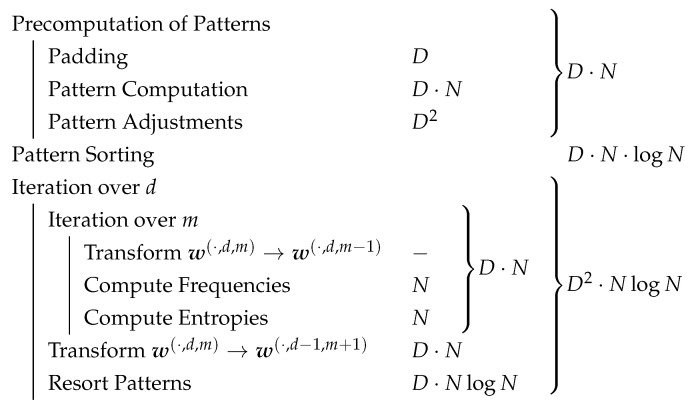
Complexity Analysis of our Algorithm, where *D* describes the maximal pattern order and *N* describes the length of the data vector.

**Figure 9 entropy-21-00547-f009:**
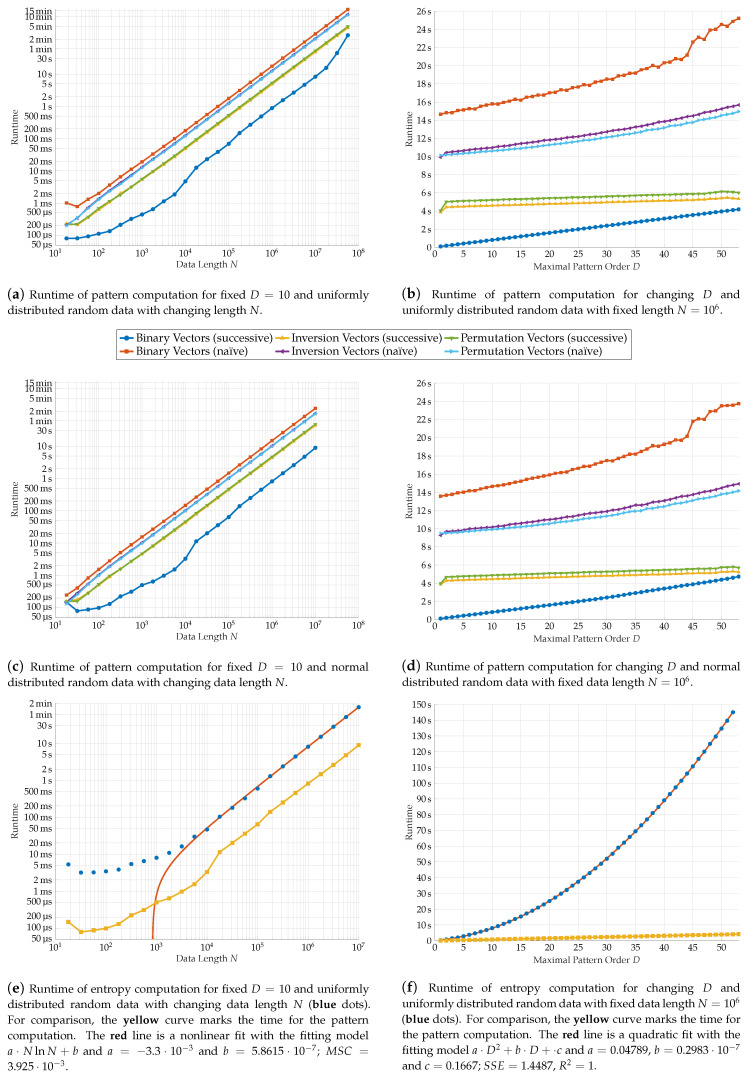
Runtime analysis of the entropy and pattern computation dependent on the data length *N* and the maximal pattern order *D*. The resulting runtimes are averaged over 50 trials.

**Figure 10 entropy-21-00547-f010:**
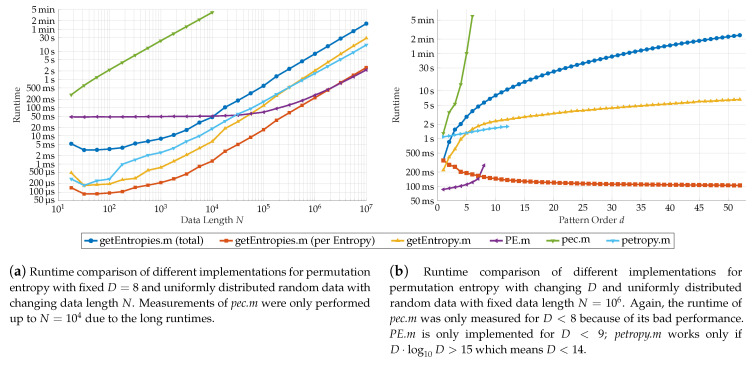
Comparision of the performance of different Matlab scripts for permutation entropy computation. The **blue** curve is the total runtime of our approach while the **red** curve gives the runtime per entropy. The modified version for a single entropy is marked **yellow**. Both are compared with the approaches from V. Unakafova (**purple**), G. Ouyang (**green**) and A. Müller (**cyan**).

**Figure 11 entropy-21-00547-f011:**
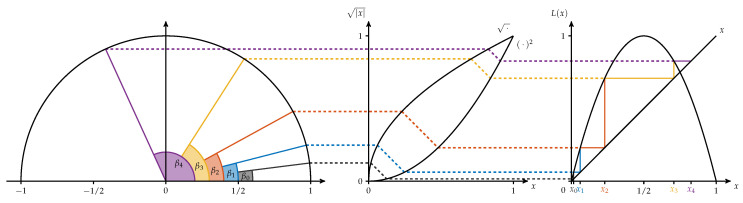
Semi-Conjugacy between the orbits of the logistic map *L* and the angle-doubling function.

**Figure 12 entropy-21-00547-f012:**
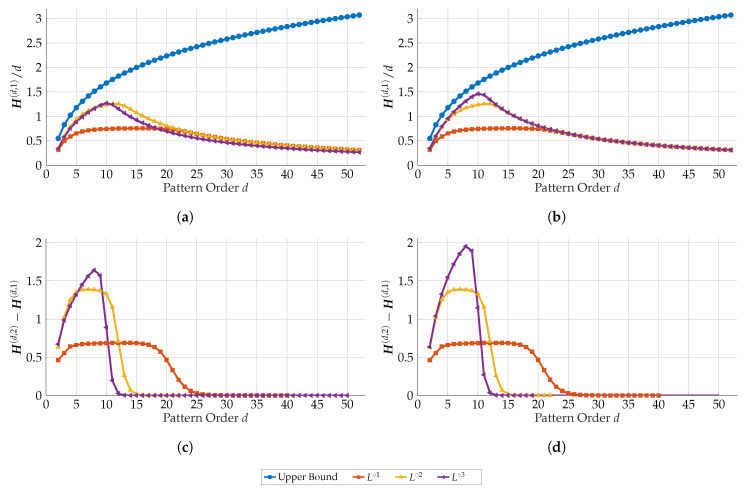
ePE (**a**,**b**) and empirical conditional entropy of ordinal patterns (**c**,**d**) for the logistic map $L=L^{\circ 1}$ and its iterates $L^{\circ 2},L^{\circ 3}$ in dependence on the pattern order *d*: Left side pictures (**a**,**c**) show results based on the naïve simulation and right side pictures (**b**,**d**) on the advanced simulation, each based on $N=10^{7}$ iterates. For comparison, in (**a**,**b**) the upper bound of the ePE (Equation 9) for *d* was added (**blue** curves).

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
