# Peer review of "Algorithmics, Possibilities and Limits of Ordinal Pattern Based Entropies"

_entropy, 2019, doi:10.3390/e21060547_

Reviewer 1 Report

The authors propose a new algorithm for computing permutation entropies. The main ideas are (1) use of a binary vector representation for ordinal patterns and (2) careful reuse of overlapping patterns. The algorithm is described in detail and its computational performance is discussed both theoretically and practically. The numerical simulation demonstrates that the run time of the proposed algorithm is comparable to that of the best existing method in small orders of ordinal patterns. Further, it can reach larger pattern orders up to 52 while keeping computational efficiency reasonable, which is impossible by any existing methods. 

The paper seems well-written. The proposed algorithm is described concisely with understandable examples and figures. Its usefulness is shown in a compelling way by comparison with existing approaches. Its application to the analysis of limitations of permutation entropies is also a nice contribution to the field of ordinal pattern analysis of time series. Thus, the reviewer recommends accepting the paper. However, the following minor comments should be considered before the publication. 

Line 85:

It seems that the order is assumed to be total as can be guessed from the claim in Line 92. If this is the case, please make the assumption explicit. 

Line 209 (above Eq.(4)):

"$l \in \{1,\dots,d\}$" --> $m \in \{ 0,\dots,d-1 \}$?

Line 216:

"word length $l$" --> word length $m$?

Line 394:

The authors wrote, "We repeated the computations several times". Please give the concrete number of trials. 

Lines 435--475 (Sec.4.4.2):

This section discusses Fig.10. However, there seems no mention within the main text. Please refer to the figure at appropriate places. 

Line 571:

"Figure (11)" --> "Figure 12"?

Line 590 (Algorithm 4):

"i -> 0" --> "i <- span="" i="" -=""> i+1" --> "i <- i+1"?

Author Response

We like to thank the reviewer for his detailed review and suggestions.

We followed his suggestions and corrected the found mistakes in our manuscript. For the second suggestion, the parameter $k$ was meant.

Reviewer 2 Report

  This manuscript presented well the scheme to hierarchically estimate permutation entropy using the ordinal patterns and the words from patterns beyond the current limitation of parameters, such as the number of data and the embedding dimension. I think this work is eligible enough fo being published.

  I want to suggest authors extend this method to assess the type II complexity of symbolic words based on ordinal symbols rather than type I complexity; permutation entropy and KS-entropy covered in the present manuscript.

Author Response

We like to thank the reviewer for his review and suggestions.

We agree with the opinion of the reviewer to consider a broader range of complexity measures, especially the measures of type II, and adopt our methods to these. They have shown in many studies to be as promising as the type I complexities we considered in the manuscript. It is indeed a future project to extend this approach in this direction.